# Adapting new norms: A mixed-method study exploring mental well-being challenges in dental technology education

**Galvin Sim Siang Lin**[1]*, **Wen Wu Tan**[2], **Kah Hoay Chua**[3], **Jong-Eun Kim**[4], **Jin Gan**[4], **Mohd Haikal Muhamad Halil**[1]

**1** Department of Restorative Dentistry, Kulliyyah of Dentistry, International Islamic University Malaysia, Kuantan Campus, Kuantan, Pahang, Malaysia, **2** Department of Dental Public Health, Faculty of Dentistry, AIMST University, Bedong, Kedah, Malaysia, **3** Department of Dental Technology, Faculty of Dentistry, AIMST University, Bedong, Kedah, Malaysia, **4** Department of Prosthodontics, Yonsei University College of Dentistry, Seoul, Korea

* galvin@iium.edu.my

## Abstract

### Background

The present study, grounded in the Stress-Adaptation-Growth theory, aims to explore the mental well-being among dental technology students during the post-pandemic period.

### Materials and methods

A mixed-method approach was employed among undergraduate dental technology students in Malaysia. The Depression, Anxiety, and Stress Scale (DASS-21) was adapted and modified for dental technology students. The content of the questionnaire was validated by two experienced faculty experts. Construct validity and internal consistency were measured. An online survey was created using Google Forms and disseminated to 10 Bachelor of Dental Technology (BDT) students. Meanwhile, qualitative data were obtained through one-on-one semi-structured interviews, employing a phenomenology approach and thematic analysis to explore students' experiences in the new educational landscape.

### Results

All students answered the survey, predominantly females. Prevalence of depression (60% normal, 10% mild, 30% moderate), anxiety (30% normal, 30% mild, 10% moderate, 10% severe, 20% extremely severe), and stress (70% normal, 10% mild, 10% moderate, 10% severe) was reported. Qualitatively, three major themes emerged: "Problems with adaptations", "Anxious about returning to campus", and "Concern about the future". The first theme included sub-themes: "Difficulties in transitioning to hybrid learning" and "Disruption in study-life balance". The second theme had four sub-themes: "Fear of being infected", "Fear of being stigmatized", "Increased vigilance in personal safety measures", and "Confusion about standard operating procedure". The third theme included sub-themes: "Fear of another lockdown" and "Concerns about timely completion of academic requirements".

**Data availability statement:** Data are available from the AIMST University Malaysia Institutional Data Access / Ethics Committee (contact via 04-4298075) for researchers who meet the criteria for access to confidential data.

**Funding:** The author(s) received no specific funding for this work.

**Competing interests:** The authors have declared that no competing interests exist.

## Conclusions

This study highlights the complex mental health challenges dental technology students faced post-pandemic, underscoring the need for flexible academic policies, global collaborations, and targeted strategies to support their resilience and well-being.

## Introduction

The COVID-19 pandemic has brought about profound and far-reaching changes in every facet of our lives, challenging our societal norms, redefining our daily routines, and reshaping our understanding of health and well-being [1]. As the world grapples with the multifaceted consequences of this global crisis, it becomes increasingly evident that its impact extends beyond the realm of physical health, leaving an indelible mark on our mental and emotional well-being [2]. In this context, the repercussions of the pandemic on diverse populations and professions have been subjected to scrutiny, illuminating the distinct challenges and stressors encountered by individuals engaged in education and careers, particularly within the health professions. Dental education bore a significant impact, being classified as high-risk due to close patient contact and the generation of aerosol droplets during dental procedures [3]. Several studies have examined the impact of COVID-19 on dental students' mental health, emphasizing its potential influence on the students' overall well-being [2,4,5].

Dental technology students, just like dental students, have faced challenges in the post-COVID-19 world. They play a pivotal role in the dental healthcare system by crafting precision dental prosthetics and managing complex laboratory equipment, may find themselves at the intersection of healthcare and technology [6], a vantage point that offers a distinctive perspective on the evolving dynamics of their field [7]. As they adapt to the demands of this new era, it is imperative to investigate and understand the impact of the pandemic on their mental well-being, an aspect often overlooked in the broader discourse on the consequences of COVID-19. While the disruptive effects of the pandemic on traditional education methods have been widely discussed [8], dental technology students' experiences have received relatively limited attention in the literature. To gain expertise in this discipline, students undergo rigorous education and training, acquiring the knowledge and skills needed to excel in the dental technology profession [9]. However, due to social distancing measures, lockdowns, and restrictions on clinical activities, dental technology students have faced challenges in gaining practical experience and hands-on training – an essential part of their curriculum. The pandemic-induced reduction in the availability of dental cases has led to a conundrum [4]; while they are required to complete a specific number of dental cases to graduate on time, the global health crisis has disrupted the supply of such cases. This predicament underscores the urgency of assessing the mental well-being of dental technology students, given the unique stressors they face in their academic journey.

Assessing the mental well-being of students is crucial for cultivating a conducive learning environment, ensuring dental technologists-to-be can navigate pandemic-induced challenges. In the current literature, many studies employed the Depression, Anxiety, and Stress Scale (DASS) as a reliable tool for gauging student psychological well-being [10–12]. However, its application in the context of dental technology students remains understudied. Complicating matters, the Asian Institute of Medicine, Science and Technology (AIMST) University is the only dental school in Malaysia, as well as the only one in the Southeast Asia region, offering the Bachelor of Dental Technology (BDT) program. This exclusivity highlights the need to evaluate the mental well-being of this select group, as their experiences may significantly differ from those in larger, more diverse educational settings. The unparalleled nature of the dental

technology program not only makes this study imperative for addressing the specific needs of this select group but also positions it as a unique opportunity to unearth insights that extend far beyond the confines of a typical dental academic setting. The findings gained from understanding the mental well-being of dental technology students can serve as a benchmark for similar programs globally, offering comparative perspectives that enrich our understanding of the challenges faced by students in specialized healthcare technology disciplines.

A comprehensive exploration of mental well-being among dental technology students in Malaysia is not only timely but also of utmost importance. It is noteworthy to emphasize that the psychological strain resulting from the challenges imposed by the pandemic and academic pressures can have far-reaching consequences for their personal and professional lives. Hence, the present qualitative study aimed to explore the lived experiences of dental technology students during the post-COVID-19 pandemic transition.

## Materials and methods

### Ethical statement

Ethical approval was obtained from the institutional human ethics committee, with the reference code AUHEC/FOD/2022/23. All students who took part in the present study provided both verbal and written consent electronically by agreeing to a consent form that was available on the Google Forms platform. The identifiable information of each student was carefully anonymized before analysis to ensure complete confidentiality. The participant information sheets explicitly communicated that participation was voluntary, interviews would be video-recorded, and students retained the right to withdraw from the study at any juncture before the data was published. Prior to the interview sessions, each participant gave their written consent, and they were afforded the chance to articulate any apprehensions or inquiries to the investigators.

### Theoretical framework

The theoretical framework underpinning this study draws on the Stress-Adaptation-Growth (SAG) theory, which posits that individuals when confronted with stressors such as a global pandemic, undergo a process of stress appraisal, adaptation, and potential growth [13]. In the context of dental technology students during the post-COVID-19 pandemic, the SAG theory offers a lens through which to understand how these students appraise and adapt to the challenges posed by disruptions in their education, practical training, and the uncertainty surrounding their future careers. The theory suggests that, in navigating the unprecedented challenges of the pandemic, students may experience varying levels of stress, but through adaptive coping mechanisms, they have the potential for personal and professional growth [14]. By applying the SAG theory, the present research aims to provide a comprehensive understanding of how dental technology students cope with the unique challenges brought about by the post-pandemic in their personal and professional lives.

### COVID-19 status during the study

On 8th March 2022, Malaysia's Prime Minister announced the country's transition to the endemic (post-pandemic) phase and the reopening of borders on 1st April 2022 [15]. Subsequently, the government eased COVID-19 restrictions, by lifting mask mandates in open spaces and allowing interstate travel regardless of vaccination status. This marked a significant shift after two years of pandemic measures. The study was conducted approximately 8 months after the transition to the endemic phase.

### Study design

This mixed method study was conducted at the Faculty of Dentistry, AIMST University, Malaysia, on 1st December 2022, where the recruitment period for the present study started from December 2022 to 20th April 2023. At the time of the study, there were only 10 BDT students for the cohort 2019-2023 throughout the entire country, and all of them were invited to take part in this study.

### Questionnaire design

The DASS-21 questionnaire used in the present study was adopted and modified to suit the context of dental technology students [16]. It consisted of 21 close-ended questions, which were categorized into three domains: (1). Depression, (2). Anxiety and (3). Stress, with a rating scale of 0 (Never), 1 (Rarely), 2 (Often) and 3 (Very Often). The content of the questionnaire was validated by two faculty experts who have extensive experience in conducting questionnaire-based research works. One of the experts holds a PhD in Dental Public Health and the other holds a Master's in Medical Education. The two experts engaged in a thorough examination and deliberation on each questionnaire item, drawing upon their collective expertise to ascertain clarity, appropriateness, and alignment with the study's objectives. Any discrepancies or differing perspectives were discussed with the investigators until a unified agreement was reached. Both experts agreed that no modifications were necessary, as the modified structure of DASS-21 adequately measured depression, anxiety, and stress among the target population.

### Quantitative data collection and analysis

An online survey was created using Google Forms and disseminated to the students via circulation of the link. Students who voluntarily agreed to participate in the study provided informed consent before answering the DASS-21 questionnaire. The construct validity is demonstrated by factor loadings of at least $\geq 0.30$ for each item using confirmatory factor analysis [17], while the internal consistency was measured post-hoc using Cronbach's alpha coefficients for each domain, with a cut-off point of 0.7 regarded as acceptable [18]. Data were analyzed using the IBM Statistical Package for the Social Sciences (SPSS) for Windows, Version 22.0. (Armonk, NY: IBM Corp., USA).

### Participants recruitment for qualitative interview

All students who completed the questionnaire were extended invitations to participate in subsequent one-on-one interview sessions, accompanied by the distribution of participant information sheets. A convenience sampling method was used and only those who were readily available and willing to participate were recruited. Nevertheless, only six individuals (out of the ten individuals) agreed to partake in the interview sessions.

### Qualitative data collection and analysis

A topic guide was created (Table 1), supported by research findings on students' learning experiences during the post-pandemic [19–21]. Due to the limited number of BDT students in Malaysia, a pre-test was conducted with five Bachelor of Dental Surgery (BDS) students, whose similar academic environments and learning experiences made them a suitable proxy. A subsequent pilot test with 10 BDS students further refined the guide's face validity, allowing for adjustments to improve clarity and usability. While the target population was BDT students, the overlap in educational challenges between BDS and BDT students during the pandemic justified this approach, ensuring the guide was well-prepared for the final interviews.

**Table 1. Interview topic guide questions.**

| Questions |
| --- |
| 1. What was your experience like on your first day back in physical classes after the MCO was lifted? |
| 2. Can you share any thoughts or feelings you had about returning to campus and resuming face-to-face activities? |
| 3. How have your interactions with classmates, faculty, or others changed since returning to campus, if at all? |
| 4. What has it been like managing both your pending pre-clinical and clinical cases along with new requirements? |
| 5. How do you feel about your academic progress and career prospects after the changes caused by the MCO? |
| 6. What challenges, if any, do you anticipate in completing your studies or preparing for your future career? |
| 7. What has it been like adapting to in-person learning and campus life again? |
| 8. Have you faced any difficulties in adjusting to the routines or expectations on campus? Can you describe them? |
| 9. How have you managed any emotions or stress since resuming on-campus activities? |
| 10. Is there anything else you'd like to share about your experience transitioning back to campus life? |

A one-on-one semi-structured interview method was selected to explore in-depth BDT students' learning experiences during the post-pandemic stage. Six interview sessions were conducted using Zoom Video Communication Software between March 2023 and June 2023, and each session lasted around 30-40 minutes. During each interview, two facilitators were present, with the first investigator leading the discussion and the second investigator providing comments. Throughout the data collection phase, the investigators met on a regular basis to share their perspectives on the recurring themes that emerged during the interview sessions. Data saturation was achieved by the fifth interview, as recurrent themes were consistently observed across participants, suggesting that additional interviews were unlikely to yield new insights. However, the sixth interview was conducted to ensure that no pertinent information was overlooked.

Data obtained from the interviews were transcribed verbatim. Following transcription, students were given the opportunity to cross-check the findings and make comments. The study employed phenomenology as its chosen methodology, with thematic analysis being utilized for data examination. The qualitative data were then analyzed using Braun and Clarke's Thematic Analysis Framework [22], which allows for a flexible yet systematic exploration of patterns and themes that emerge from the data. This approach is well-suited for identifying recurring themes without being bound to a rigid theoretical structure. First, two investigators (WWT and GSSL) familiarized themselves with the data by repeatedly reading the transcripts to identify recurring patterns and key insights, followed by performing preliminary coding using NVIVO 12 software. These codes were then grouped into preliminary themes and sub-themes, ensuring that related concepts were categorized effectively. The themes were subsequently reviewed and refined, with investigators cross-checking them against the dataset. Any discrepancies in the codes were resolved by the third investigator (KHC). The final set of codes was collectively agreed upon by the entire study team.

## Results

### Quantitative findings

All 10 eligible students (response rate of 100%) answered the questionnaire survey, with the majority being females (90%) and the remaining being males (10%). Most (60%) of the students were aged between 21 and 22, while 20% were aged 23 to 24, and another 20% were aged 25 to 26. Cronbach's alpha values for depression (0.81), anxiety (0.84), and stress (0.90) indicated acceptable reliability. Moreover, the confirmatory factor analysis demonstrated an acceptable value of 0.9. Based on the Shapiro-Wilk normality test, the data were normally

distributed across all domains: Depression (W = 0.943, p = 0.482), Anxiety (W = 0.967, p = 0.763), and Stress (W = 0.921, p = 0.305).

Table 2 highlights the prevalence of depression, anxiety, and stress among dental technology students in the post-COVID-19 era. In general, most students experienced normal levels of depression (60%), anxiety (30%), and stress (70%). Moderate depression was reported by 30%, while anxiety levels varied, with 30% experiencing severe to extremely severe symptoms. No cases of extremely severe depression and stress were noted.

Table 3 reveals the mean and standard deviation for each questionnaire item. The first domain (Item 1 to Item 7) evaluated students' depression level with the highest mean score noted in Item 2 whereby most students found it challenging to work up the initiative to start and complete tasks in the dental laboratory. Nevertheless, most of them never feel that life is meaningless as a dental technology student. The second domain (Item 8 to Item 14) evaluated students' anxiety level where most of the students worried about situations in the dental laboratory where they might feel panicked or make mistakes. Meanwhile, the majority of the students never feel scared without any good reason while working in the dental laboratory. Domain 3 (Item 15 to Item 21) evaluated students' stress levels with the highest mean score found in Item 17 where most students felt that they used a lot of nervous energy while working on dental laboratory work.

## Qualitative findings

Three main themes were identified namely, "Adaptation problems", "Anxious about returning to the campus" and "Concern about future", as illustrated in Fig 1.

**Theme 1: Problems with adaptations.** *Sub-theme 1: Difficulties in transitioning to hybrid learning*: Students faced numerous challenges in the learning process when the COVID-19 transitional period was announced. Apart from online classes, they had to travel to the campus for their practical sessions.

**Table 2. Prevalence of depression, anxiety, and stress among Bachelor of Dental Technology (BDT) students in times of post-COVID-19 era.**

| DASS Level | Sample, n (%) |
|---|---|
| **Depression** | |
| Normal | 6 (60%) |
| Mild | 1 (10%) |
| Moderate | 3 (30%) |
| Severe | 0 (0%) |
| Extremely severe | 0 (0%) |
| **Anxiety** | |
| Normal | 3 (30%) |
| Mild | 3 (30%) |
| Moderate | 1 (10%) |
| Severe | 1 (10%) |
| Extremely severe | 2 (20%) |
| **Stress** | |
| Normal | 7 (70%) |
| Mild | 1 (10%) |
| Moderate | 1 (10%) |
| Severe | 1 (10%) |
| Extremely severe | 0 (0%) |

**Table 3. Descriptive results (mean and standard deviation) of students' responses to the questionnaire.**

| Questions | | Mean (s.d.) |
|---|---|---|
| **Depression** | | |
| 1. | I struggled to experience any positive feelings as a dental technology student. | 0.9 (0.74) |
| 2. | I found it challenging to work up the initiative to start and complete tasks in the dental laboratory. | 1.0 (0.67) |
| 3. | I felt that I had nothing to look forward to as a dental technology student. | 0.5 (0.53) |
| 4. | I felt downhearted and blue while working on dental laboratory work. | 0.7 (0.67) |
| 5. | I could not seem to become enthusiastic about anything related to my dental laboratory work. | 0.9 (0.74) |
| 6. | I did not feel like I had much self-worth as a dental technology student. | 0.4 (0.70) |
| 7. | I felt that life was meaningless as a dental technology student | 0.2 (0.42) |
| **Anxiety** | | |
| 8. | I noticed dryness of my mouth while working in the dental laboratory. | 1.1 (0.99) |
| 9. | I experienced difficulty in breathing, such as rapid breathing or breathlessness, even without physical exertion, during my dental laboratory work. | 0.4 (0.52) |
| 10. | I experienced trembling, particularly in my hands, while working on delicate dental laboratory work. | 0.9 (0.88) |
| 11. | I was worried about situations in the dental laboratory where I might feel panicked or make mistakes. | 1.5 (0.97) |
| 12. | I felt I was close to panic while working on complex dental laboratory work. | 0.8 (0.79) |
| 13. | I was aware of my heart rate increasing or having irregular beats, in the absence of physical exertion, while working on dental laboratory work. | 0.5 (0.53) |
| 14. | I felt scared without any good reason while working in the dental laboratory. | 0.3 (0.67) |
| **Stress** | | |
| 15. | I found it challenging to unwind after a busy day in the dental laboratory. | 1.0 (1.15) |
| 16. | I tended to overreact to unexpected situations in my dental laboratory work. | 0.8 (0.63) |
| 17. | I felt that I used a lot of nervous energy while working on dental laboratory work. | 1.1 (0.74) |
| 18. | I found myself getting agitated when dealing with complex dental laboratory cases. | 0.8 (0.63) |
| 19. | I found it difficult to relax during my study. | 0.7 (0.67) |
| 20. | I became intolerant of anything that disrupted my workflow in the dental laboratory. | 1.0 (0.67) |
| 21. | I felt rather touchy when receiving feedback on my dental laboratory work. | 0.6 (0.70) |

s.d.: standard deviation.

P2: *"We need to rush back and forth to the hostel and to the lab… it is tiring sometimes."*

P3: *"Rushing from hostel and then to lab and then from lab back to hostel again for online classes."*

Persistent technical problems with online classes also posed a significant challenge to the students.

P1: *"Wi-Fi in the hostel is not stable and we cannot hear the lecturer clearly."*

P2: *"The lecturers' Wi-Fi is lagging."*

P6: *"The Wi-Fi in our hostel is very weak. Sometimes we cannot follow the classes well."*

**Sub-theme 2: Disruption in study-life balance:** Students felt stressed and found it difficult to adjust as they were already used to staying at home throughout the pandemic. They also experienced an increase in academic workload because the practical sessions were resumed.

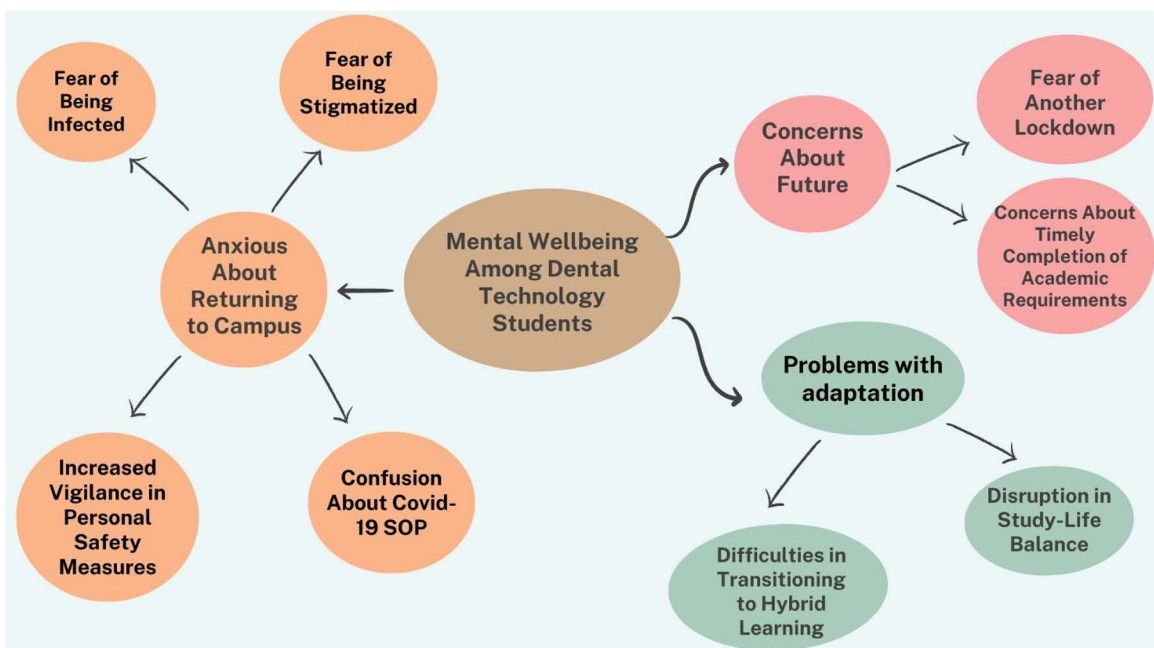

**Fig 1. Thematic map.** Themes and sub-themes highlighted by dental technology students.

*P1: "I feel stress since I have been used to staying at home for the online classes. It is difficult to achieve study life balance."*

*P3: "After the endemic, I must come back to campus. There are lots of assignments.".*

*P4: "I feel stressed because a lot of work to do."*

*P5: "It is a bit difficult because too much of work. I cannot complete my tasks on time."*

**Theme 2: Anxious about returning to the campus.** *Sub-theme 1: Fear of being infected*: Most students were concerned about returning to dental school because the number of cases was still high at the time. They were concerned about becoming infected with the virus.

*P1: "I am anxious because we are in an enclosed space in the lab and might be infected."*

*P3: "I am worried that I may be infected."*

*Sub-theme 2: Fear of being stigmatized*: Students were also concerned about being stigmatized by others if they became infected with COVID-19.

*P5: "I am also worried that if I get infected, they will talk bad about me."*

*P5: "My roommate once infected by COVID-19 and many people accused her."*

*Sub-theme 3: Increased vigilance in personal safety measures*: Many of them took extra precautions to ensure that they stayed safe during the endemic. They would also stay away or advise any symptomatic individuals to be examined.

*P2: "I will ensure that I wear mask even though the government announced endemic."*

*P3: "I will try to keep my distance away from people who is coughing."*

*Sub-theme 4: Confusion about standard operating procedure (SOP):* The constantly changing COVID-19 SOP also caused confusion among the students, particularly regarding the quarantine period for confirmed cases and close contact cases.

P4: *"It makes me confused. If I am close COVID-19 contact, the class might be affected."*

P6: *"Sometimes we are confused about how many days do we need to quarantine if infected."*

**Theme 3: Concern about future.** *Sub-theme 1: Fear of another lockdown:* Students were concerned that another lockdown would be imposed if the number of reported COVID-19 cases increased.

P2: *"I also feel worried that if there is another lockdown."*

P3: *"I am afraid there will be a sudden movement control order and I need to go back home immediately."*

*Sub-theme 2: Concerns about timely completion of academic requirements:* Many feared that they would not be able to complete their cases in time should there be another lockdown in the future.

P3: *"I scared I cannot finish my quota (requirements) on time, and I have to delay until next semester."*

P4: *"I feel stressful and still trying my best to catch up with all my quota."*

## Discussion

The present study assessed the mental well-being among dental technology students in the post-pandemic era yielding both quantitative and qualitative insights. The convergence of quantitative and qualitative findings signifies the multidimensional nature of mental health and well-being challenges faced by dental technology students, providing a comprehensive picture of the psychosocial landscape post-COVID-19. The high internal consistency observed in the reliability assessment of the questionnaire items, as indicated by Cronbach's alpha values exceeding the 0.7 threshold, lends credibility to the findings. In addition, the acceptable values in confirmatory factor analysis further support the validity of the measurement tool, affirming the appropriateness of the chosen domains for assessing mental well-being among dental technology students. It is noteworthy that most of the students reported normal levels across both the depression and stress domains with half of them claiming normal to mild anxiety levels, while approximately half of them reported experiencing normal to mild levels of anxiety. This trend suggests a prevailing sense of resilience among dental technology students.

Based on the current results, most students (60%) did not encounter depression following the post-pandemic era upon resuming university activities, with 80% of them expressing that they never perceived life as meaningless as a dental technology student. These findings align with a previous study conducted by Lestari *et al.* (2022) reporting a mere 9.1% constant prevalence of depression among Malaysian dental students [2]. This concurrence may be attributed to heightened awareness of COVID-19 infection, the accessibility of vaccines, and the relaxation of lockdown measures within the country during the investigation period [23,24]. The present findings also align with a prior study conducted among Italian students, which similarly demonstrated a rapid reduction in depressive symptoms following the easing of lockdown measures [24]. Nevertheless, it is important to highlight that some students found it challenging to work up the initiative to start and complete their tasks in the dental laboratory.

This suggests that certain individuals might experience persistent post-lockdown depression, highlighting the potential long-term effects of the pandemic on mental health [25].

The presence of mild to moderate anxiety among dental technology students suggests that the practical aspects of working in the dental laboratory may induce apprehension in some individuals. Meanwhile, the identification of severe and extremely severe anxiety among students highlights a notable subset grappling with heightened emotional distress. This can be linked to the qualitative findings where current dental technology students expressed anxiety about returning to campus due to fears of infection and the potential stigma associated with it as reported in a similar study [3]. Such worry and anxiety among dental technology students may account from numerous incidents of stigmatization against healthcare workers and COVID-19 patients worldwide [26,27]. Thus, to address social stigma, one should show empathy to affected individuals, understand the disease itself and speak up against negative stereotypes. The elevated anxiety levels may also stem from various factors, including the demanding and intricate nature of laboratory tasks, the fear of making errors, and the pressure associated with practical sessions [28]. This corroborates with the high mean score in Item 11, where many students expressed concern about feeling panicked or making mistakes in the dental laboratory. Indeed, an essential component of the dental technology program involves hands-on practical sessions, allowing students to proficiently develop the expertise required to create perfect dental prostheses [9]. It is conceivable that the intricate and precise nature of laboratory tasks, combined with the expectation of accuracy and precision in the dental field, creates a heightened sense of anxiety and pressure among students [29]. The authors postulated that the COVID-19 lockdown has significantly reduced these practical chances, increasing anxiety among students about potential blunders when returning to hands-on work after a long absence.

On a positive note, most students reported not feeling scared without any valid reason while working in the dental laboratory. Similarly, most students reported experiencing normal stress levels. This observation may reflect a resilient aspect of the students' coping mechanisms [30], indicating that, despite experiencing the transition from COVID-19 pandemic to endemic phase, they maintain a rational and composed approach to their coursework and practical responsibilities in the dental laboratory. Moreover, none of the students reported extremely severe stress suggests that, despite the challenges posed by dental laboratory work, most students have not reached a level of stress that significantly impairs their ability to cope. This points towards the effectiveness of the educational support system in place, but ongoing monitoring and tailored interventions may be beneficial to further mitigate stressors and promote optimal mental health among dental technology students [31]. It is also crucial to recognize the significance of normal stress levels in a demanding academic and laboratory setting, as a certain degree of stress can serve as a motivational factor [32], driving students to meet the rigors of their dental technology education. Nevertheless, these identified anxieties and stresses underscore the need to foster a supportive learning environment, address specific stressors in the dental laboratory, and integrate mental health awareness into the dental technology curriculum to enhance overall student well-being.

The current findings revealed that dental technology students struggled with adaptability issues during the post-pandemic period. Many of them were accustomed to the flexibility of online lessons and found it difficult to adjust as they now had to travel to the laboratory to attend their practical sessions. Similar struggles were observed among international students studying in the United Kingdom, who were reluctant to give up on the flexibility in learning after experiencing online teaching [33]. Apart from that, most students were anxious about returning to the campus due to the fear of being infected and confusion about the COVID-19 protocol. Similar concerns were seen in past studies [21,34], and only 32.6% of the students

were ready to return to the campus upon reopening [34]. As a result, universities need to adopt clear and strict COVID-19 guidelines during this crucial period to ensure the optimal health and well-being of students as they transit back from online to physical learning.

Furthermore, students expressed varied degrees of concern about their future, encompassing worries about meeting laboratory requirements within stipulated timeframes and the potential recurrence of lockdowns. The results were consistent with the previous studies conducted among undergraduate dental students where a shared concern was identified regarding the timely completion of their dental degrees and a preference not to extend the duration of their studies [2,21]. The anxiety and stress surrounding the completion of quota requirements and potential delays due to unforeseen circumstances underscores the need for flexible academic policies and support structures. Several studies have echoed the current finding that students harbor apprehensions regarding the potential resurgence of COVID-19 [35,36]. This alignment in perspectives may be attributed to the persisting uncertainties surrounding the virus, with the ongoing emergence of novel variants adding to the complex landscape of understanding. The continuous evolution of our knowledge about the virus contributes to a sense of unease among students, as the uncertainties surrounding new variants, and their implications remain prominent.

Limitations of the present study include its cross-sectional design as it does not allow the assessment of the dynamic changes in the psychological state of dental technology students over time. This design restricts the ability to assess whether anxiety, depression, and stress levels increased, decreased, or remained stable during different phases of the pre-and post-pandemic. Secondly, the relatively small sample size may limit the depth and diversity of perspectives. While the internal consistency of the questionnaire items yielded acceptable Cronbach's alpha values (all exceeding the 0.7 threshold), these results should be interpreted with caution due to the small sample size. The confirmatory factor analysis also demonstrated acceptable values; however, the limited participant pool restricts the generalizability of the results. As the sole dental school providing a bachelor's degree in dental technology across the entire country and Southeast Asia region, the limited availability of comparable institutions makes it challenging to draw direct comparisons with findings from other similar studies. Furthermore, the use of self-report measures, such as the DASS-21 questionnaire, relies on participants' subjective responses, which may be influenced by social desirability or recall biases. The study did not exclude students who had received psychological treatment, potentially introducing bias as their responses may be influenced by the treatment received. Additionally, the study focuses on a specific cohort of dental technology students in Malaysia. Given the unique characteristics of this population, caution should be exercised when generalizing the findings to dental technology students in different countries.

In addressing the unique challenges faced by dental technology students post-COVID-19, recommendations for dental technology education include targeted interventions for practical aspects, such as virtual reality integration. This is essential to support students in adapting to hybrid learning environments and managing the practical demands of laboratory work. Academic institutions should also invest in stable online platforms, provide training for hybrid learning, and ensure technical support for both students and faculty. Mental health support is crucial, requiring proactive programs, counselling services, and stress management workshops to create a supportive environment and diminish the stigma around seeking help. Addressing the psychological impact of the pandemic and promoting overall well-being are crucial for fostering resilience among students. To further safeguard student progress, institutions must adopt flexible academic policies that include contingency plans for sudden transitions between in-person and online learning. This adaptability will help ensure that students can meet academic and practical requirements even amid unforeseen challenges. Future research

should focus on long-term mental health effects through longitudinal studies and global comparisons among dental technology programs to better understand the broader implications of pandemic-related disruptions. By addressing these issues, institutions can enhance the resilience and mental well-being of dental technology students, contributing to a more robust and supportive educational landscape in the evolving post-pandemic era.

## Conclusion

In conclusion, this study provides a comprehensive exploration of the impact of the post-pandemic on the mental well-being of dental technology students in Malaysia, an aspect often overlooked in the broader discourse on the consequences of the global crisis. The study, grounded in the Stress-Adaptation-Growth (SAG) theory, reveals that students experienced varying levels of stress, anxiety, and depression during the post-COVID-19 era. While most students experienced normal to mild levels of stress and depression, a notable proportion reported moderate to severe anxiety. The qualitative insights further highlighted adaptation problems, anxiety about returning to campus, and concerns about the future as prominent themes. These findings underscore the need for targeted support and interventions to address the mental well-being of dental technology students, considering their unique position at the intersection of healthcare and technology. As the world transitions into the endemic phase of the pandemic, it is crucial to prioritize the psychological well-being of these students, recognizing the potential long-term consequences on both personal and professional aspects of their lives. Future research and institutional initiatives should further explore effective strategies for supporting dental technology students in overcoming challenges posed by the post-COVID-19 landscape and fostering their resilience and growth in the face of adversity.

## Author contributions

**Conceptualization:** Galvin Sim Siang Lin, Wen Wu Tan.

**Data curation:** Galvin Sim Siang Lin, Wen Wu Tan, Kah Hoay Chua.

**Formal analysis:** Galvin Sim Siang Lin, Wen Wu Tan, Kah Hoay Chua.

**Investigation:** Kah Hoay Chua, Jin Gan, Mohd Haikal Muhamad Halil.

**Methodology:** Galvin Sim Siang Lin, Wen Wu Tan, Kah Hoay Chua.

**Project administration:** Galvin Sim Siang Lin.

**Software:** Kah Hoay Chua, Jong-Eun Kim.

**Validation:** Jong-Eun Kim, Jin Gan, Mohd Haikal Muhamad Halil.

**Visualization:** Wen Wu Tan.

**Writing – original draft:** Galvin Sim Siang Lin, Wen Wu Tan, Kah Hoay Chua.

**Writing – review & editing:** Jong-Eun Kim, Jin Gan, Mohd Haikal Muhamad Halil.

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
