## [Decision Letter · Decision Letter 0]

1 Oct 2024

PONE-D-24-32730Adapting New Norms: A Mixed-Method Study Exploring Mental Well-being Challenges in Dental Technology EducationPLOS ONE

Dear Dr. Lin,

Thank you for submitting your manuscript to PLOS ONE. After careful consideration, we feel that it has merit but does not fully meet PLOS ONE’s publication criteria as it currently stands. Therefore, we invite you to submit a revised version of the manuscript that addresses the points raised during the review process.

We look forward to receiving your revised manuscript.

Kind regards,

Vanessa Carels

Staff Editor

PLOS ONE

Journal Requirements:

Reviewers' comments:

Reviewer's Responses to Questions

**Comments to the Author**

1. Is the manuscript technically sound, and do the data support the conclusions?

Reviewer #1: Partly

Reviewer #2: Partly

2. Has the statistical analysis been performed appropriately and rigorously?

Reviewer #1: Yes

Reviewer #2: No

3. Have the authors made all data underlying the findings in their manuscript fully available?

Reviewer #1: No

Reviewer #2: No

4. Is the manuscript presented in an intelligible fashion and written in standard English?

Reviewer #1: Yes

Reviewer #2: Yes

5. Review Comments to the Author

Reviewer #1: Please attach semi structured interview guide.

Please elaborate the validity of the guide.

Why phenomenological approach was employed?

Please align conclusions with the findings of the study.

Please review the themes and subthemes and consider presenting in a tabular form if feasible.

Please mention the framework utilised for qualitative analysis

Reviewer #2: Thank you for the great effort in this article. It seems you have spent an enormous amount of time conducting this study. The main advantage of such study is its unique population.

Here are some comments.

Abstract:

Mention the number of participants and response rate if available.

Clarify the aim of the qualitative part.

In the conclusion, report findings related to the levels of depression, anxiety, and stress.

Rephrase "content validation" as "content was validated."

Introduction:

Provide a citation for the claim "may find themselves at the intersection of healthcare and technology."

Clarify whether "dental technology students" refers only to Malaysia or if it applies globally.

Add citations for "Challenges in the hybrid learning method" and "Change in study-life balance."

The introduction has some redundancy and could be more concise. Highlight studies that have investigated the psychological health of dental technologists. For example, consider including studies like this one to establish previous research in this area.

Methods:

The "COVID-19 Status During the Study" section should be shortened. Include only the study period in brief. This section reads as though it was derived from a thesis, so be more concise and adapt it to fit a journal article.

"At the time of the study conducted, there were only 10 BDT students for the cohort 2019-2023 throughout the entire country" – Do you mean there are only 10 students in this specialty across all cohorts? Please clarify.

Explain what modifications were made to the questionnaire. The DASS-21 is widely used without modification. Clarify why changes were necessary.

Instead of "faculty experts," specify their specialties and qualifications (e.g., Master's, PhD).

For a sample size of 10 students, using factor loadings and Cronbach's alpha might not be meaningful due to the small sample size. Consider reporting only descriptive statistics.

Participant recruitment for qualitative interviews should be made more concise.

When referring to "WWT" and "GSSL," define them the first time you mention them (provide abbreviations for the investigators).

Mention the sampling method (e.g., purposeful sampling).

Results:

Summarize the quantitative results briefly, and if reporting Cronbach's alpha, do so without exaggerating the findings (e.g., "indicate robust reliability across the questionnaire items" is too strong for such a small sample).

Remove Table 1 and present the data in the text, as there are only a few variables.

Avoid redundancy in reporting the results. For example, rather than repeating text from Table 2, simply state, "The prevalence of depression, anxiety, and stress among BDT students is shown in Table 2."

The extensive changes made to the DASS-21 may affect whether the study is measuring depression, anxiety, and stress as intended, or whether it is measuring aspects more related to the profession. I recommend focusing on measuring depression, anxiety, and stress, and leaving the contextual analysis to the qualitative part.

Indicate whether the data were normally distributed; otherwise, use median and IQR.

Avoid redundancy throughout the article, such as repeating "Only six students agreed to take part in the interview sessions," which has been mentioned earlier.

The qualitative data analysis should be more in-depth than the quantitative part, as it is critical to the study's findings.

Discussion:

Avoid over-exaggerating the validity and consistency of the questionnaire. You can discuss these aspects if you conducted a pilot with a larger sample (e.g., 12 participants) before using the questionnaire in your main study.

In the second paragraph, clarify whether the Malaysian and Italian studies you referenced involved dental students or dental technology students specifically.

Reduce repetition in the manuscript. For example, the implications and recommendations for dental education are repeated in different paragraphs. Combine these into one cohesive section to avoid redundancy.

6. PLOS authors have the option to publish the peer review history of their article (what does this mean? ). If published, this will include your full peer review and any attached files.

**Do you want your identity to be public for this peer review?** For information about this choice, including consent withdrawal, please see our Privacy Policy .

Reviewer #1: **Yes: ** Malik Zain Ul Abideen

Reviewer #2: No

---

## [Author Response · Author response to Decision Letter 1]

26 Dec 2024

Reviewer 1

1. Please attach semi structured interview guide.

Reply: Thank you for your valuable feedback. We appreciate your suggestion regarding the inclusion of the semi-structured interview guide. We have now added the interview guide in Table 1 for your reference.

Page 9:

“Table 1. Interview topic guide questions”

2. Please elaborate the validity of the guide.

Reply: Thank you for your thoughtful feedback regarding the validity of the interview guide. We have elaborated on this aspect by including the following sentences in the manuscript.

Page 8:

"Due to the limited number of BDT students in Malaysia, a pre-test was conducted with five Bachelor of Dental Surgery (BDS) students, whose similar academic environments and learning experiences made them a suitable proxy. A subsequent pilot test with 10 BDS students further refined the guide’s face validity, allowing for adjustments to improve clarity and usability. While the target population was BDT students, the overlap in educational challenges between BDS and BDT students during the pandemic justified this approach, ensuring the guide was well-prepared for the final interviews."

We hope this addition clarifies the steps taken to ensure the validity of the interview guide. Thank you for your valuable input and consideration.

3. Why phenomenological approach was employed?

Reply: Phenomenology was employed in this study because it focuses on investigating and comprehending individuals' lived experiences. Given the aim to explore students' emotional and personal experiences during the COVID-19 pandemic, this approach allowed for a deeper understanding of how they perceived and navigated academic challenges. Phenomenology is well-suited to capture the richness and complexity of these experiences, providing valuable insights into the unique ways students adapted to and overcame obstacles in their learning journey.

4. Please align conclusions with the findings of the study.

Reply: We have modified the conclusion as suggested by both reviewers.

5. Please review the themes and subthemes and consider presenting in a tabular form if feasible

Reply: Thank you for your valuable feedback regarding the presentation of themes and sub-themes. We have carefully reviewed and revised the themes and sub-themes. This adjustment enhances the clarity and readability of the qualitative findings, while preserving the original meaning and integrity of each theme and sub-theme. As Figure 1 effectively conveys the necessary information, we believe that adding a table would result in redundancy. Nonetheless, we appreciate your insightful suggestion, which has contributed to improving the overall presentation of the manuscript.

6. Please mention the framework utilised for qualitative analysis

Reply: Thank you for your insightful comment. We would like to clarify that the qualitative analysis in this study was guided by Braun and Clarke’s Thematic Analysis Framework. This approach was selected for its flexibility and ability to identify, analyze, and report patterns directly from the data, allowing for a rich, detailed exploration of students’ experiences without being restricted by a rigid theoretical model.

We have added two sentences to the methodology section.

Page 9:

“The qualitative data were then analyzed using Braun and Clarke’s Thematic Analysis Framework, which allows for a flexible yet systematic exploration of patterns and themes that emerge from the data. This approach is well-suited for identifying recurring themes without being bound to a rigid theoretical structure.”

Reviewer 2

1. Thank you for the great effort in this article. It seems you have spent an enormous amount of time conducting this study. The main advantage of such study is its unique population.

Reply: Thank you for highlighting the uniqueness of the study population. We believe this focus on dental technology students will contribute valuable insights to the field of allied health profession.

2. Mention the number of participants and response rate if available.

Reply: We have addressed this by stating the number of participants in the results section.

Page 10:

"All 10 eligible students (100% response rate) …."

3. Clarify the aim of the qualitative part.

Reply: We have addressed your comment by adding the aim of the qualitative study to the last paragraph of the introduction section.

Page 5:

“Hence, the present qualitative study aimed to explore the lived experiences of dental technology students during the post-COVID-19 pandemic transition.”

4. In the conclusion, report findings related to the levels of depression, anxiety, and stress.

Reply: We have refined the conclusion by adding the general findings.

Page 21:

“While most students experienced normal to mild levels of stress and depression, a notable proportion reported moderate to severe anxiety.”

5. Rephrase "content validation" as "content was validated”.

Reply: We have rephrased the sentence.

Page 7:

“The content of the questionnaire was validated by two faculty experts…”

6. Introduction:

Provide a citation for the claim "may find themselves at the intersection of healthcare and technology."

Reply: Thank you for your insightful feedback. We have added a citation from a previous qualitative study we conducted among dental technologists to substantiate this statement.

We acknowledge the limited literature available on dental technologists' perceptions, and this has been highlighted within the manuscript. We hope this study contributes to increasing awareness among readers and healthcare professionals about the importance of addressing the welfare of allied dental professionals.

Comment: Clarify whether "dental technology students" refers only to Malaysia or if it applies globally.

Reply: We would like to clarify that this study specifically focuses on dental technology students in Malaysia. However, as mentioned in the introduction, we believe that the findings can serve as a benchmark for similar programs globally, providing valuable comparative perspectives.

Comment: Add citations for "Challenges in the hybrid learning method" and "Change in study-life balance."

Reply: The phrases "Challenges in the hybrid learning method" and "Change in study-life balance" are actually sub-themes derived from our qualitative findings, as detailed in the Results section. However, we appreciate your attention to this point, and we would like to highlight that these sub-themes are further elaborated in the Discussion section, where appropriate references and relevant literature are cited to support and contextualise our findings.

Comment: The introduction has some redundancy and could be more concise. Highlight studies that have investigated the psychological health of dental technologists. For example, consider including studies like this one to establish previous research in this area.

Reply: We greatly appreciate your suggestion to highlight studies on the psychological health of dental technologist students. However, after conducting an extensive literature search across PubMed, Scopus, and Google Scholar using various keywords [("dental technologist" OR "dental technician") AND ("mental health" OR "psychological health" OR "well-being" OR "stress" OR "anxiety" OR "depression")], we found very limited studies focusing specifically on depression, anxiety or stress among dental technologist students or dental technicians. This gap in the literature highlights the novelty and importance of our study.

In addition, Malaysia is the only country in Southeast Asia offering a Bachelor's degree in Dental Technology, further highlighting the uniqueness of this population and the challenges in drawing comparisons or citing similar studies. We believe this study contributes to addressing this gap and provides valuable insights for future research in this underrepresented area.

7. Methods:

Comment: The "COVID-19 Status During the Study" section should be shortened. Include only the study period in brief. This section reads as though it was derived from a thesis, so be more concise and adapt it to fit a journal article.

Reply: Thank you for your valuable feedback on the "COVID-19 Status During the Study" section. We agree with your point that a more concise version aligns better with the format of a journal article. However, we would like to note that previous reviewers recommended elaborating on this aspect to provide a clearer understanding of the context and circumstances during the study period.

Nevertheless, we have taken your suggestion into account and have summarised this section.

Page 6:

“On 8th March 2022, Malaysia's Prime Minister announced the country's transition to the endemic (post-pandemic) phase and the reopening of borders on 1st April 2022 ….. The study was conducted approximately 8 months after the transition to the endemic phase.”

Comment: "At the time of the study conducted, there were only 10 BDT students for the cohort 2019-2023 throughout the entire country" – Do you mean there are only 10 students in this specialty across all cohorts? Please clarify.

Reply: At the time the study was conducted, there were only 10 Bachelor of Dental Technology (BDT) students for the 2019-2023 cohort across the entire country. The BDT program at AIMST University is the only undergraduate dental technology program in Malaysia, with an annual intake of fewer than 10 students. This highlights the rarity and significance of this group.

Furthermore, across Southeast Asia, this is the only bachelor-level dental technology program available, further emphasising the exclusivity of this program and its unique position within the region. We believe that investigating the mental health and academic challenges faced by this distinct and underrepresented group will provide valuable insights that could inform educational strategies and policies, not just in Malaysia but potentially for similar programs globally.

Comment: Explain what modifications were made to the questionnaire. The DASS-21 is widely used without modification. Clarify why changes were necessary.

Reply: Thank you for your insightful comment regarding the modifications made to the DASS-21 questionnaire. We appreciate your attention to this detail.

In the manuscript, we mentioned that the DASS-21 was adapted and modified to suit the context of dental technology students. As shown in Table 3, general terms of each questionnaire item were adjusted to explicitly refer to "dental technology students" to enhance clarity and applicability within this unique cohort.

We believe these contextual adjustments preserve the integrity of the original scale while ensuring the questionnaire reflects the experiences of our target population.

Comment: Instead of "faculty experts," specify their specialties and qualifications (e.g., Master's, PhD).

Reply: We have revised the manuscript to provide greater clarity regarding the qualifications and specialties of the experts involved in the validation process.

Comment: For a sample size of 10 students, using factor loadings and Cronbach's alpha might not be meaningful due to the small sample size. Consider reporting only descriptive statistics.

Reply: Regarding the appropriateness of factor loadings and Cronbach’s alpha, we completely understand the concern about the limited statistical power associated with a sample of 10 students. However, during an earlier phase of the review process, previous reviewers recommended including factor analysis and Cronbach’s alpha to demonstrate the internal consistency and validity of our adapted questionnaire, even with the small sample. In addition, we consulted our university’s biostatistician, who advised us to proceed with these analyses to provide an overview of the psychometric properties of the instrument. We acknowledge the limitations associated with small-sample factor analysis and have therefore interpreted the results with caution. We appreciate your understanding and are open to further guidance if necessary.

Comment: Participant recruitment for qualitative interviews should be made more concise. Mention the sampling method (e.g., purposeful sampling).

Reply: We have revised the participant recruitment section, by adding phrase indicating ‘one-on-one’ interviews and the sampling method.

Page 8:

“A convenience sampling method was used and only those who were readily available and willing to participate were recruited.”

Comment: When referring to "WWT" and "GSSL," define them the first time you mention them (provide abbreviations for the investigators).

Reply: To ensure investigator anonymity, we have revised the manuscript to refer to "GSSL" and "WWT" as the first and second investigators, respectively.

8. Results:

Comment: Summarise the quantitative results briefly, and if reporting Cronbach's alpha, do so without exaggerating the findings (e.g., "indicate robust reliability across the questionnaire items" is too strong for such a small sample).

Reply: We appreciate your suggestion regarding the presentation of the quantitative results. In response, we have revised the sentence.

Page 10:

“Cronbach’s alpha values for depression (0.81), anxiety (0.84), and stress (0.90) indicated acceptable reliability.”

Comment: Remove Table 1 and present the data in the text, as there are only a few variables.

Reply: We have removed Table 1 as suggested and have presented the demographic data within the text.

Page 10:

"Most (60%) of the students were aged between 21 and 22, while 20% were aged 23 to 24, and another 20% were aged 25 to 26."

Comment: Avoid redundancy in reporting the results. For example, rather than repeating text from Table 2, simply state, "The prevalence of depression, anxiety, and stress among BDT students is shown in Table 2."

Reply: We appreciate your suggestion regarding reducing redundancy in reporting the results. In response, we have summarized the paragraph.

Page 10:

"Most students experienced normal levels of depression (60%), anxiety (30%), and stress (70%). Moderate depression was reported by 30%, while anxiety levels varied, with 30% experiencing severe to extremely severe symptoms. No cases of extremely severe depression and stress were noted."

This revision reduces word count while maintaining clarity and conciseness.

Comment: The extensive changes made to the DASS-21 may affect whether the study is measuring depression, anxiety, and stress as intended, or whether it is measuring aspects more related to the profession. I recommend focusing on measuring depression, anxiety, and stress, and leaving the contextual analysis to the qualitative part.

Reply: We would like to clarify that the modifications made to the DASS-21 were minimal and did not alter the core content or structure of the instrument. The changes primarily involved adapting the terminology to reflect the context of dental technology students, ensuring relevance without deviating from the original meaning. To maintain the integrity of the scale, the modified version underwent content validation by two experts with experience in dental education and psychometric evaluation. This step ensured that the questionnaire remains a reliable measure of depression, anxiety, and stress as originally intended. We appreciate your attention to detail and trust this clarification addresses your concern.

Comment: Indicate whether the data were normally distributed; otherwise, use median and IQR

Reply: Thank you for your valuable feedback regarding the data distribution. We have conducted a normality test (Shapiro-Wilk) for each domain of the DASS-21 scores. The results indicate that the data were normally distributed. Given that the p-values for all domains exceed 0.05, we have proceeded with reporting the mean and standard deviation for the quantitative results.

Page 10:

“Based on the Shapiro-Wilk normality test, the data were normally distributed across all domains: Depression (W=0.943, p=0.482), Anxiety (W=0.967, p=0.763), and Stress (W=0.921, p=0.305).”

Comment: Avoid redundancy throughout the article, such as repeating "Only six students agreed to take part in the interview sessions," which has

---

## [Decision Letter · Decision Letter 1]

29 Jan 2025

PONE-D-24-32730R1Adapting New Norms: A Mixed-Method Study Exploring Mental Well-being Challenges in Dental Technology EducationPLOS ONE

Dear Dr. Lin,

Thank you for submitting your manuscript to PLOS ONE. After careful consideration, we feel that it has merit but does not fully meet PLOS ONE’s publication criteria as it currently stands. Therefore, we invite you to submit a revised version of the manuscript that addresses the points raised during the review process.

We look forward to receiving your revised manuscript.

Kind regards,

Ayesha Fahim

Academic Editor

PLOS ONE

Journal Requirements:

Reviewers' comments:

Reviewer's Responses to Questions

**Comments to the Author**

1. If the authors have adequately addressed your comments raised in a previous round of review and you feel that this manuscript is now acceptable for publication, you may indicate that here to bypass the “Comments to the Author” section, enter your conflict of interest statement in the “Confidential to Editor” section, and submit your "Accept" recommendation.

Reviewer #3: (No Response)

Reviewer #4: All comments have been addressed

2. Is the manuscript technically sound, and do the data support the conclusions?

Reviewer #3: Yes

Reviewer #4: Yes

3. Has the statistical analysis been performed appropriately and rigorously?

Reviewer #3: Yes

Reviewer #4: Yes

4. Have the authors made all data underlying the findings in their manuscript fully available?

Reviewer #3: Yes

Reviewer #4: Yes

5. Is the manuscript presented in an intelligible fashion and written in standard English?

Reviewer #3: Yes

Reviewer #4: Yes

6. Review Comments to the Author

Reviewer #3: The authors have significantly improved the manuscript after revision as per reviewers' suggestions and comments. There are few suggestions for further improvement.

1. Abstract: Rephrase “These 21 close-ended questionnaire items were content validation.”

2. There are two styles of using apostrophe comma which should be consistent throughout the manuscript.

3. In subheading of Participants Recruitment for Qualitative Interview, the following statement should be moved to Ethics consideration to avoid redundancy. “The participant information sheets explicitly communicated that participation was voluntary, interviews would be video recorded, and students retained the right to withdraw from the study at any juncture before the data was published. Prior to the interview sessions, each participant gave their written consent, and they were afforded the chance to articulate any apprehensions or inquiries to the investigators.”

4. In discussion, 2nd paragraph, better to move the reference [2] at the end of the sentence.

Reviewer #4: To improve the provided text, the abstract should explicitly outline the study’s objectives and include the exact sample size for transparency. Grammatical corrections, such as changing “among undergraduate” to “among undergraduates,” are necessary for clarity. Additionally, the abstract should conclude with a clear statement highlighting the practical implications of the findings. In the introduction, the literature review needs to be expanded to include more studies on mental health challenges in dental technology students, and the choice of population and setting should be justified.

The materials and methods section should provide detailed information about the modifications made to the DASS-21 tool, including its validation process for this specific population. The sampling technique for interviews needs to be explained thoroughly, specifying how participants were selected and when data saturation was achieved. Furthermore, the thematic analysis requires a more detailed description of the steps followed to ensure transparency. In the results section, quantitative findings should specify sample sizes and include visual aids such as charts or graphs to enhance understanding. For qualitative findings, participant quotes should be included to support identified themes, and more detailed demographic information should be provided.

The discussion section would benefit from a comparative analysis with similar studies to contextualize the findings and from addressing potential biases, such as those arising from self-reporting, along with limitations in data collection. The conclusion should emphasize the practical applications of the findings to strengthen its relevance to academic or clinical settings. References must adhere to the PLOS ONE format, and recent studies should be added to enhance the study’s credibility.

Across the manuscript, grammatical errors and typographical issues need to be corrected, and the language should maintain a consistent academic tone. Adding visual representations for quantitative data and summary tables for qualitative themes will improve clarity and accessibility for readers. A clear data availability statement should be included to ensure transparency and allow replication or verification of the study’s results. Overall, improving methodological transparency, ethical clarity, and reporting of qualitative rigor will significantly enhance the study’s credibility and alignment with PLOS ONE standards.

7. PLOS authors have the option to publish the peer review history of their article (what does this mean? ). If published, this will include your full peer review and any attached files.

**Do you want your identity to be public for this peer review?** For information about this choice, including consent withdrawal, please see our Privacy Policy .

Reviewer #3: No

Reviewer #4: No

---

## [Author Response · Author response to Decision Letter 2]

7 Feb 2025

Editor

Response: We sincerely appreciate your feedback regarding the reference list. We have carefully checked each reference and confirm that none of the cited papers have been retracted.

In addition, we have added the page numbers to four references to ensure completeness and accuracy (References 20, 25, 26 and 33).

We have ensured that our reference list adheres to PLOS ONE formatting guidelines (We used EndNote for PlosOne). Thank you for your attention to this matter, and we appreciate your valuable insights.

Reviewer 3

1. The authors have significantly improved the manuscript after revision as per reviewers' suggestions and comments. There are few suggestions for further improvement. Abstract: Rephrase “These 21 close-ended questionnaire items were content validation.”

Response: The author has revised the abstract.

Page 2:

“The content of the questionnaire was validated by two experienced faculty experts.”

2. There are two styles of using apostrophe comma which should be consistent throughout the manuscript.

Response: We appreciate the reviewer's attention to detail. We have carefully reviewed the manuscript and standardized the use of apostrophes for consistency throughout the text.

3. In subheading of Participants Recruitment for Qualitative Interview, the following statement should be moved to Ethics consideration to avoid redundancy. “The participant information sheets explicitly communicated that participation was voluntary, interviews would be video recorded, and students retained the right to withdraw from the study at any juncture before the data was published. Prior to the interview sessions, each participant gave their written consent, and they were afforded the chance to articulate any apprehensions or inquiries to the investigators.”

Response: Thank you for your valuable feedback. We have addressed your suggestion by shifting the statement from the "Participants Recruitment for Qualitative Interview" section to the "Ethics Consideration" section.

4. In discussion, 2nd paragraph, better to move the reference [2] at the end of the sentence.

Response: Thank you for your valuable feedback. We have revised the second paragraph in the Discussion section by moving reference [2] to the end of the sentence, as suggested. This improves the clarity and readability of the text.

Reviewer 4

1. To improve the provided text, the abstract should explicitly outline the study’s objectives and include the exact sample size for transparency. Grammatical corrections, such as changing “among undergraduate” to “among undergraduates,” are necessary for clarity. Additionally, the abstract should conclude with a clear statement highlighting the practical implications of the findings. In the introduction, the literature review needs to be expanded to include more studies on mental health challenges in dental technology students, and the choice of population and setting should be justified.

Response: We sincerely appreciate your time and effort in reviewing our manuscript and for your constructive feedback. Below are our responses to your comments.

In response, we have revised our abstract by including the sample size.

“…10 Bachelor of Dental Technology (BDT) students.”

Regarding the proposed revision ("Among Undergraduate" to "Among Undergraduates"), we respectfully would like to maintain the phrase "among undergraduate dental technology students".

We also highlighted the significance of our study for academic policies, student well-being, and global collaborations at the end of the abstract.

Regarding the comment on expanding the literature review and justification of population and setting, we would like to humbly clarify that, as stated in our response to the first and second reviewers, there are no existing studies focusing on the mental health challenges of undergraduate dental technology students during the COVID-19 pandemic and post-pandemic era. To ensure a comprehensive literature review, we also conducted a literature search on 5 February 2025 using Web of Science, Scopus, and PubMed databases. Despite our extensive search, we found no published studies specifically addressing this population. The closest available literature pertains to dental students, which we have duly cited and referenced in our introduction to provide relevant contextual support.

Furthermore, the selection of dental technology students as our study population has been justified in our introduction. This population represents a unique subset of health professions education, with distinct academic and clinical training challenges, especially in the wake of pandemic-related disruptions. Given that AIMST University is the only institution in Malaysia and Southeast Asia offering a Bachelor of Dental Technology (BDT) program, our study holds significant value in addressing the mental health challenges of a uniquely specialized student group, whose experiences may differ from those of dental students and other health profession students.

We hope this clarifies our approach and justifications. Once again, we appreciate your valuable feedback and are happy to make further refinements if needed.

2. The materials and methods section should provide detailed information about the modifications made to the DASS-21 tool, including its validation process for this specific population. The sampling technique for interviews needs to be explained thoroughly, specifying how participants were selected and when data saturation was achieved. Furthermore, the thematic analysis requires a more detailed description of the steps followed to ensure transparency.

Response: We sincerely appreciate your thorough review and valuable feedback on our manuscript. Below are our responses to your comments:

In our Materials and Methods section, we have already described how the questionnaire was adapted and modified to suit the context of dental technology students. We explained the content validation process, which was conducted by two faculty experts, one with a PhD in Dental Public Health and another with a Master's in Medical Education. The validation process involved a thorough examination and deliberation on each questionnaire item to ensure clarity, appropriateness, and alignment with the study's objectives. Any discrepancies were discussed and resolved until consensus was reached. To further enhance transparency, we added a sentence.

Page 7:

“Both experts agreed that no modifications were necessary, as the modified structure of DASS-21 adequately measured depression, anxiety, and stress among the target population.”

Moreover, we mentioned that the study employed convenience sampling, where all students who completed the survey were invited to participate in interviews. Due to the limited number of undergraduate dental technology students in Malaysia, only six students agreed to participate in the interviews. Nonetheless, we observed recurrent themes in our interviews, suggesting that additional interviews were unlikely to yield new insights. We have added 2 sentences to improve clarity.

Page 9:

“Data saturation was achieved by the fifth interview, as recurrent themes were consistently observed across participants, suggesting that additional interviews were unlikely to yield new insights. However, the sixth interview was conducted to ensure that no pertinent information was overlooked.”

We appreciate the suggestion to provide a more detailed description of the thematic analysis process. Our study followed Braun and Clarke’s Thematic Analysis Framework. To enhance transparency, we expanded our description of the thematic analysis steps to ensure that readers fully understand our approach.

Page 10:

“First, two investigators (WWT and GSSL) familiarized themselves with the data by repeatedly reading the transcripts to identify recurring patterns and key insights, followed by performing preliminary coding using NVIVO 12 software. These codes were then grouped into preliminary themes and sub-themes, ensuring that related concepts were categorized effectively. The themes were subsequently reviewed and refined, with investigators cross-checking them against the dataset.”

3. In the results section, quantitative findings should specify sample sizes and include visual aids such as charts or graphs to enhance understanding. For qualitative findings, participant quotes should be included to support identified themes, and more detailed demographic information should be provided.

Response: We sincerely appreciate your thoughtful feedback. We would like to clarify that in our Results section, we have explicitly stated that all 10 eligible students (100% response rate) participated in the study. Additionally, Table 2 and Table 3 provide a detailed breakdown of the prevalence of depression, anxiety, and stress among students, as well as the mean and standard deviation for each questionnaire item. Furthermore, we have incorporated visual aids to illustrate the key findings, summarizing the identified qualitative themes and sub-themes. We hope this addresses the concern regarding the inclusion of sample sizes and visual aids to enhance understanding.

We would also like to highlight that direct participant quotes are already included in our Results section, supporting each identified theme and sub-theme. Furthermore, we have specified key demographic details of our participants, including gender distribution (90% female, 10% male) and age groups (21-26 years old), to ensure a clear understanding of our study population.

4. The discussion section would benefit from a comparative analysis with similar studies to contextualize the findings and from addressing potential biases, such as those arising from self-reporting, along with limitations in data collection. The conclusion should emphasize the practical applications of the findings to strengthen its relevance to academic or clinical settings. References must adhere to the PLOS ONE format, and recent studies should be added to enhance the study’s credibility.

Response: We sincerely appreciate your constructive feedback.

We acknowledge the importance of contextualizing our findings by comparing them with similar studies. However, as noted in our manuscript, our study focuses on dental technology students, a unique and underrepresented population, with AIMST University being the only institution in Malaysia and Southeast Asia offering a Bachelor’s degree in Dental Technology. Due to the limited availability of comparable studies, drawing direct comparisons is inherently challenging.

Nevertheless, we have cited relevant studies on dental students, as they share some overlapping academic stressors and challenges. In our revision, we did incorporating literature on mental health among health profession students, particularly those in hands-on based disciplines.

We also appreciate the recommendation to further discuss potential biases in our study. In our manuscript, we have already acknowledged the possible biases stemming from:

• Self-report measures (DASS-21) – Participants' responses may have been influenced by social desirability bias or recall bias.

• Psychological treatment history – The study did not exclude students who may have received mental health interventions, which could influence their responses.

• Small sample size – Due to the limited population of dental technology students, findings should be interpreted with caution.

We acknowledge the reviewer's recommendation to emphasize the practical applications of our findings. Our conclusion did outlined key interventions such as:

“These findings underscore the need for targeted support and interventions to address the mental well-being of dental technology students, considering their unique position at the intersection of healthcare and technology. As the world transitions into the endemic phase of the pandemic, it is crucial to prioritize the psychological well-being of these students, recognizing the potential long-term consequences on both personal and professional aspects of their lives. Future research and institutional initiatives should further explore effective strategies for supporting dental technology students in overcoming challenges posed by the post-COVID-19 landscape and fostering their resilience and growth in the face of adversity.”

5. Across the manuscript, grammatical errors and typographical issues need to be corrected, and the language should maintain a consistent academic tone. Adding visual representations for quantitative data and summary tables for qualitative themes will improve clarity and accessibility for readers. A clear data availability statement should be included to ensure transparency and allow replication or verification of the study’s results. Overall, improving methodological transparency, ethical clarity, and reporting of qualitative rigor will significantly enhance the study’s credibility and alignment with PLOS ONE standards.

Response: We sincerely appreciate your detailed feedback and suggestions for improving the manuscript. We reviewed the manuscript to correct any grammatical and typographical issues.

We have already included tables and a figure summarizing both quantitative data and qualitative themes, which we believe are sufficient for clarity.

A clear data availability statement has been incorporated (in our cover letter) to ensure transparency.

Thank you for your valuable insights, and we appreciate your time and effort in reviewing our manuscript.

---

## [Editor Report · Decision Letter 2]

21 Feb 2025

Adapting New Norms: A Mixed-Method Study Exploring Mental Well-being Challenges in Dental Technology Education

PONE-D-24-32730R2

Dear Dr. Lin,

We’re pleased to inform you that your manuscript has been judged scientifically suitable for publication and will be formally accepted for publication once it meets all outstanding technical requirements.

Kind regards,

Ayesha Fahim

Academic Editor

PLOS ONE
---

## [Editor Report · Acceptance letter]

PONE-D-24-32730R2

PLOS ONE

Dear Dr. Lin,

I'm pleased to inform you that your manuscript has been deemed suitable for publication in PLOS ONE. Congratulations! Your manuscript is now being handed over to our production team.

Kind regards,

on behalf of

Dr. Ayesha Fahim

Academic Editor

PLOS ONE